# Ageing and Long-Term Informal Care: The Reality of Two Countries in Europe: Denmark and Portugal

**DOI:** 10.3390/ijerph191710859

**Published:** 2022-08-31

**Authors:** Ana Catarina Maia, Paulo Nogueira, Maria Adriana Henriques, Carla Farinha, Andreia Costa

**Affiliations:** 1Health Sciences Research Unit: Nursing (UICISA: E), Nursing School of Coimbra (ESEnfC), 3000-076 Coimbra, Portugal; 2Nursing Research, Innovation and Development Centre of Lisbon (CIDNUR), Nursing School of Lisbon, 1600-190 Lisbon, Portugal; 3Instituto de Saúde Ambiental (ISAMB), Faculdade de Medicina, Universidade de Lisbon, 1350-300 Lisbon, Portugal; 4Center for Environmental and Sustainability Research (CENSE), NOVA School of Science and Technology (FCT), NOVA University of Lisbon, 1099-085 Lisbon, Portugal

**Keywords:** ageing, caregivers, Denmark, health data, long-term informal care, Portugal

## Abstract

The knowledge of long-term informal care is particularly interesting for social and health measures related to ageing. This study aims to analyze how Portugal differs from Denmark regarding long-term informal care, specifically referring to personal care received by older people. A cross-sectional study was developed in Portugal and Denmark through the Survey of Health, Ageing and Retirement in Europe (SHARE) in 2015, with a total of 2891 participants. Descriptive statistics and logistic regressions were performed. The findings suggest a significant association for older people from Portugal who receive long-term informal care from non-household caregivers and household caregivers. Moreover, as they age and are from Portugal, their availability to receive long-term informal care from non-household caregivers increases. Furthermore, older people in Portugal are more likely to receive long-term informal care from a household caregiver. It is important to take a closer look at long-term informal care in both countries and think about healthy ageing policies in the current context of the ageing population. This study provides knowledge about disaggregated health data on ageing in the European region, helping to fill research gaps related to older people.

## 1. Introduction

In Europe today, we are witnessing an ageing population with an increasing trend for the coming years, which is associated with a growth in long-term informal care for older people [1].

Recently, there has been a proposed definition of long-term informal care, which is assumed to involve care provided to people who need support because of disability or old age and refers to care activities that may involve help with activities of daily living (e.g., bathing, dressing, and eating) and instrumental activities of daily living (e.g., shopping, meal preparation, housework or administration) [2]. Similarly, long-term informal care is assumed to be provided for three consecutive months by someone in the social environment of the care recipient (for example, a family member, friend or neighbor), and the caregiver is not paid for providing care [2,3].

The Portuguese sociodemographic reality demonstrates the European trend of the increase in the population’s ageing index, especially in recent years [4]. Ageing is associated with a high rate of total dependence, especially concerning dependence for activities of daily living (ADLs) and instrumental activities of daily living (IADLs), with repercussions in the scope of long-term informal care [5].

The population ageing index (ratio of “population aged 65 years or older” to “population aged 15 years or younger”) in Portugal was 153.2% in 2017, revealing a growth trend in recent years [6,7]. The Danish context, in turn, verifies the trend of other European countries, with the ageing rate fixed at 112.9% in 2017 and the older people dependency rate (ratio of “population of 65 years or over” and “population between 15 and 64 years”) fixed at 29.6% [8].

Recent data have shown that in the 4th and 6th waves of SHARE, Portugal had the highest percentage of people older than 50 years provided with informal care within the household in both waves, namely, 11.5% in the 4th wave and 12.7% in the 6th wave compared to other countries [9]. Nevertheless, there is an inverse relationship with the rate of coverage of formal care, whose reasons point to the scarcity of formal caregivers, the insufficient socioeconomic resources of informal caregivers, and the cultural issue of family care [9,10,11].

On the other hand, in Denmark, studies point to a lower provision of long-term informal care compared to formal care, which is a situation justified by the better socioeconomic conditions of older people and their families, as well as by the social and health policies instituted in health care [12,13].

A recent report by Organization for Economic Co-operation and Development (OECD) shows that although these two countries in 2017 presented different contexts in comparison to OECD countries, the percentage of informal caregivers older than 50 years was on average 13%, and a wide variation in the intensity of care provided was evident, namely, in Portugal (8.2%) and Denmark (5.3%) [1]. These data demonstrate that Denmark is one of the countries with the lowest rates of daily care, similar to other Nordic countries, whose situation is also associated with the well-developed long-term formal care sector and better-charged coverage [1,14].

In Portugal, the reality is different, showing that most informal caregivers provide daily and more intensive care [15]. Portugal is the second OECD country with the highest rate of informal day-care provided by women (70.1%), leading to an associated burden with care and negative effects on the health of caregivers, particularly females [1,15].

On the other hand, the overall way in which long-term care is implemented in both countries reveals different realities, with an impact on informal care [16].

In the Danish context, long-term care consists of four types of preventative measures, rehabilitation, home care, and homes for older people, and is implemented publicly or privately by the municipalities, mainly free of charge and financed by general taxation, with the aim of deinstitutionalization, rehabilitation, and autonomy, where the role of informal caregivers tends to be smaller compared to other European countries [14,16].

In the Portuguese case, the formal system of long-term care, besides being traditionally operated by social organizations, includes the national network of integrated long-term care with a social and health component and a high rate of utilization, which raises questions about its financial sustainability [16]. Moreover, problems of access and price affordability concerning formal long-term care persist, reflected in one of the highest rates of care provided by informal caregivers in the EU-27 member states [15].

Given the above, this study is of fundamental social importance to understand and analyze the characteristics of long-term informal care in the two countries.

The general aim of the present study is to analyze how different Portugal is from Denmark regarding long-term care. Two specific goals will be addressed:(a)How is long-term informal care received by older people in both countries, concerning personal care by informal caregivers who are part of their household?(b)How is long-term informal care received by older people in both countries, concerning personal care by informal caregivers who do not live with them?

It is expected that the present study will identify differences and similarities between the two countries in the context of older people being beneficiaries of long-term informal care.

In addition, the data from this study will allow a better understanding of long-term informal care in both countries, showing the importance of the evolution of knowledge and the need to conduct studies related to the study, especially in the face of the current pandemic situation for Corona Virus Disease (COVID-19).

Indeed, recent studies already show that during this period, informal caregivers have demonstrated numerous difficulties related to (a) concern for their well-being and that of the older people they care for and the consequent increase in the burden related to mental health illnesses (isolation and fear of contracting the virus); (b) socioeconomic factors (loss of employment) and; (c) difficulties in accessing health care for themselves and the people they care for [17,18].

## 2. Materials and Methods

This study is cross-sectional, observational, and analytical. We used SHARE as the basis for our data study. SHARE is the first longitudinal and cross-national research project collecting data on ageing and provides access to the informal care sector of people older than 50 years in 27 countries in Europe and Israel [19]. Additional information about the survey can be found on the project’s webpage [20].

### 2.1. Sample and Data

The present study used data from the SHARE 6th wave, release 6.1.1, that occurred in 2015 [21,22]. The SHARE’s 6th wave used probability-based sampling (drawn either by simple random selection or multistage random selection) and was conducted using computer-assisted personal interviews (CAPI) in panel households [21]. For the 6th wave of SHARE, probability-based sampling was used to draw inferences about the population aged 50 years and older by a multi-stage stratified sampling design [23].

For the Portuguese SHARE sample, the target population was defined as all Portuguese-speaking residents, born until 1960, and their spouses/partners, regardless of their age [24]. For the Danish SHARE sample, the target population was defined as all residents who speak Danish and who were born until 1960 [24].

In both countries, data collection was conducted through interviews, according to the questionnaires available for wave 6, either in Portuguese or Danish [25].

In both samples, analysis of the width and magnitude of the 95% confidence interval (95% CI) was used to evaluate the uncertainty of estimates.

Considering the purpose of the present study, one additional restriction was applied to the target population in both countries, namely: 65 years or older. Thus, we adopted the definition of older people proposed by the European Union, as a person who is 65 years old [26]. In the case of Denmark, the final sample size corresponded to 1878 respondents, and, in Portugal, the final sample size was 1013 respondents, corresponding to a total number of participants of 2891 respondents.

### 2.2. Measures of Variables

In this study, the dependent variables were (a) long-term informal care, referring to personal care, received by older people from non-household caregivers, and (b) long-term informal care, referring to personal care received by older people from household caregivers.

For the variable “long-term informal care, referring to personal care, received by older people from non-household caregivers”, we considered all who answered affirmatively to the question: “*Please look at card 27. Thinking about the last twelve months, has any family member from outside the household, any friend or neighbor given you any kind of help listed on this card?*” (SP002) and cumulatively chose the type of help “*personal care: i.e., dressing, bathing or showering, eating, getting in or out of bed, using the bath*” in the question “*Which types of help has this person provided in the last twelve months?*” (SP004). For further details, see Appendix A and Appendix B.

For the variable “long-term informal care received by older people from household caregivers”, we considered all who answered affirmatively to the question: “*And is there someone living in this household who has helped you regularly during the last twelve months with personal care, such as washing, getting out of bed, or dressing?*” (SP020). For further details, see Appendix C.

Moreover, in the characterization of long-term informal care received by older people from both non-household and household caregivers, regarding personal care, the kinship of the caregivers was considered. As such, the responses to the questions: “*Which (other) family member, friend, or neighbor helped you in the last twelve months?*” (SP003) and “*Who helps you with personal care in the household?*” (SP021) were considered. Hence, the variable “kinship” was created, which resulted from the aggregation of the responses to the answer: “*Who gave you help?*” into three categories: “spouse/partner”; “child”; and “friends/other relatives/neighbors”. It is also important to highlight that in the construct of the “kinship” variable, the grouping into the referred categories resulted only from the options chosen by the older people who composed our sample. For further information, see Appendix A and Appendix C.

The frequency of long-term informal care received by the older people from caregivers outside the household was also considered in the characterization and resulted from the response to the question: “*In the last twelve months how often altogether have you received such help from this person? Was it…*?” (SP005). In this way, the variable “frequency” was created from the aggregation of the answers in three categories: “*almost every day*”; “*almost every week*” and “*almost every month/less often*”. Regarding the frequency of long-term informal care received by the older people from household caregivers, it is important to mention that in the wave 6 questionnaire, in the question: “*And is there someone living in this household who has helped you regularly during the last twelve months with personal care, such as washing, getting out of bed, or dressing?*” (SP020), the authors informed that “*By regularly we mean daily or almost daily during at least three months*”. No further information about the frequency of this care was asked throughout the questionnaire.

Older people, from both countries, who simultaneously reported receiving help with personal care from both non-household and household caregivers constituted a small group (*N* = 24) and were therefore excluded from the analysis.

Considering the literature review, two groups of variables of interest were considered in the study analysis, namely: (a) sociodemographic and economic variables and (b) health variables.

The sociodemographic variables considered were age at the time of the interview (in 2015) and further categorization into six age categories; gender; marital status; educational level (categorized according to ISCED 97 [27]; and country. The economic status and annual household income were considered (variable “*thinc*”) to categorise income terciles.

The health variables that were used in the study referred to the presence of chronic diseases, limitation in ADLs, limitation in IADLs and depression, and binary variables were created from the variables “*chronic*”, “*adl*”, “*iadl*”, and “*Euro-D*”.

For the classification of functional dependence, the following measurement instruments were used: the Katz index to characterize ADLs and the Lawton and Brody index to characterize IADLs [27]. Depression was assessed using the Euro-D scale [27].

Table 1 presents the study’s variables considered to characterize older people and long-term informal care.

### 2.3. Statistical Analysis

This study was developed in two phases. In the first phase, chi-square tests (χ^2^) were performed to assess the differences between the older people in the two countries under study, namely, Denmark and Portugal, in terms of their socio-demographic, economic, and health factors, as well as in the characteristics of the long-term informal care received, referring to personal care. In addition, effect size measures (Phi/Cramer’s *V*), proposed by Cohen, and their confidence intervals were used to identify statistically significant results, even if small differences existed [28].

Calibrated individual weights were used in this descriptive analysis due to the non-uniform sample design of the SHARE survey.

In the second stage, multivariate logistic regressions were conducted to assess the determinants for the occurrence of the two types of long-term care received by older people about personal care, i.e., those provided by non-household informal caregivers or by household informal caregivers. For both types of long-term informal care, sociodemographic, economic, and health variables were considered (Model 1). Finally, an interaction (Model 2) was introduced to determine whether age categories change older people’s availability to receive formal long-term care, considering their country.

The likelihood ratio test was used to assess the significance, the Hosmer and Lemeshow test was used to assess the fit of the models, and residual analyses were performed. The “Enter” method was applied for each model.

For all statistical analyses, a *p*-value < 0.05 was chosen as the significance level.

## 3. Results

### 3.1. Sociodemographic, Economic, and Health Participants’ Characteristics

The sociodemographic, economic, and health characteristics of the study population according to the country are shown in Table 2.

The statistical tests for the comparison of the two groups showed significant differences for all analyzed variables between the group of older people from Denmark and that from Portugal.

Overall, the Portuguese older people group was older than the Danish older people group (83.0 years compared to 77.7 years).

In addition, the older people group in Portugal had a lower percentage in the following age categories: 65–69 years old (28.6% compared to 33.5% in the Danish group), 70–74 years old (25.2% compared to 25.6% in the Danish group), and ≥90 years old (2.2% compared to 4.0% in the Danish group). On the other hand, it was found that the older people group in Portugal had a higher percentage in the 75–79 age group (17.5%, compared to 17.1% in the Danish group), 80–84 age group (17.3%, compared to 12.5% in the Danish group), and 95–89 age group (9.2% in the Portuguese group compared to 7.3% in the Danish group). As for gender, we observed that in both countries the majority of participants were women, especially in Portugal (58.3% compared to 54.0% in Denmark). The majority of the older people in both countries were married or living with a partner, especially those of Portuguese nationality (72.3% compared to 65.0% in Denmark). In terms of the level of education, the older people group from Denmark had higher levels of education than the older people group from Portugal (40.8% and 36.0% codified with “3” and “5” considering ISCED 97 compared to 63.4% of older people placed in ISCED-code “1” in the Portuguese group). Regarding annual household income, the older people group from Denmark had a higher household income than the older people group from Portugal (43.5% with high income compared to 71.0% with low income in Portugal). In terms of health, the older people group from Portugal had the highest proportion of participants who reported the presence of more than two chronic diseases (74.0% compared to 55.4% of the Danes); the highest proportion of older people with depression (48.7% compared to 39.8% of the Danes); the highest proportion of participants with one or more limitations in ADLs (18.8%, compared with 7.0% of the Danes; and the highest proportion of participants with one or more than one limitation in IADLs (23.8% compared with 15.4% of the Danes).

Considering the effect size, which measures the magnitude of the differences found, they were significant for most of the variables under analysis, except for the gender and depression (Euro-D). Therefore, large effect sizes were identified in the variables referring educational level and household income; medium effect sizes were identified in the marital status variable; and small effect sizes were identified in the remaining variables, namely, age (years) and age (classes), chronic disease, depression; limitations in ADLs; limitations in IADLs.

### 3.2. Long-Term Informal Care Characteristics Received by Older People Relative to Personal Care

The characterization of the long-term informal care received, relative to personal care by the study population, according to the country, is shown in Table 3.

In terms of the long-term informal care received by older people from non-household caregivers, we observed statistically significant differences between the two groups regarding the frequency of its occurrence; when it is provided by the children, by friends/other relatives/neighbors; and if it is provided almost every day, almost every week, and almost every month/less often. Overall, the Portuguese older people group had a higher percentage of this type of care received (5.6% compared to 2.7% in Denmark), a higher percentage of care provided by children (3.2% compared to 1.6% in Denmark) and by friends/other relatives/neighbor’s (2.7% compared to 1.7 in Denmark); and a higher percentage of care received almost daily (5.3% compared to 1.5% in Denmark). Nevertheless, the Danish group of older people showed a higher percentage when the frequency of care occurred almost every week (1.3% compared to 0.4% of the Portuguese group of older people) and when it occurred almost every month/less often (1.4% compared to 0.6% of the Portuguese group).

Regarding the long-term informal care received by older people, relative to personal care, from household caregivers, there were statistically significant differences between the groups regarding the occurrence of care and whether it was provided by either spouses or children. The older people group in Portugal had a higher percentage of long-term informal care provided by informal caregivers (12.6% compared to 2.1% in Denmark), a higher percentage of care provided by their spouse (4.8% compared to 2.2% in Denmark) and by children (5.3% compared to 0.1% in Denmark). The effect size on the variables described above was found to be significant, although of small magnitude.

### 3.3. Ability of Older People to Receive Long-Term Informal Care, Related to Personal Care

To explain the ability of older people to receive long-term informal care, related to personal care, either by non-household caregivers or household caregivers, logistic regressions were performed (Table 4).

Regarding the ability of older people to receive informal care, related to personal care, by caregivers outside the household, model 1 explained about 21.8% of the variation in the receipt of this care and indicated age, marital status, household income, number of chronic diseases, limitations in ADLs and IADLs as significant variables.

Indeed, according to model 1, the older people were, the more susceptible they were to receiving care from a caregiver outside the household (OR = 1.04), as well as older people not living with their spouse (OR = 1.82). Moreover, older people with a higher annual household income had lower odds of receiving informal care from caregivers outside the household (OR = 0.22). Considering health, those who were more likely to receive informal care, related to personal care, were all those who reported having more than one or two chronic diseases (OR = 2.19), one or more limitations in ADLs (OR = 3.08), and one or more limitations in IADLs (OR = 2.90).

To understand whether age changed the association between the country of the older people and the likelihood of receiving informal care from caregivers outside the household, an interaction term (country*age) was included.

Model 2, which explained about 22.5% of the variance in older people receiving long-term informal care from non-family carers, showed that the results remained stable except for the results for age and country. Hence, it was observed that the older people from Portugal and the country*age interaction was significantly associated with receiving informal care from caregivers outside the household (OR = 1.07).

On the other hand, regarding the likelihood of older people receiving informal care, referring to personal care, from household caregivers, model 1 explained 39.3% of this care, with age, marital status, level of education, annual household income, country, depression, and limitations in ADLs and IADLs as the variables considered significant. Therefore, according to model 2, the probability of receiving informal care from caregivers belonging to their household was highly correlated with the age of the older people (OR = 1.08), similar to older people from Portugal (OR = 7.35). Older people with the highest educational level (OR = 2.02); annual household income (OR = 2.23); depression (OR = 1.60); and limited ADL (OR = 6.18) and IAVDS (OR = 8.09) were more likely to be cared for by caregivers in the household. On the other hand, older people who did not live with their spouses were less likely (OR = 0.17) to be cared for by a household caregiver.

To understand whether age changed the association between the country of the older people and the likelihood of receiving informal care provided by household caregivers, an interaction term (country*age) was introduced.

Thus, model 2, which explained 39.5% of the variation in the older people receiving long-term informal care from household caregivers, showed that the results remained stable except for the country. Indeed, whether the older people were from Portugal and were more aged were not significantly associated with the fact that they received informal care by caregivers belonging to the household.

## 4. Discussion

Long-term informal care for older people has increasingly become a phenomenon worth studying within health and social sciences due to its impact on society and health policymaking [13,29].

The present study focuses on analyzing the relationship between the long-term informal care received by older people, regarding personal care provided by caregivers from outside or within their household, and the country of their origin, namely, Portugal and Denmark.

The results of our study suggested that there is a significant association between whether older people receive long-term informal care, concerning personal care, from caregivers outside the household or from household caregivers, and their country. Reinforcing the aspects mentioned above, this study also shows that older people from Portugal are more likely to receive long-term informal care, concerning personal care, from caregivers belonging to the household and, as they get older, they are more likely to receive these types of care.

These results are in line with some studies conducted in recent years, which show that in Portugal the provision of long-term informal care of older people who need support in their personal care, either by household or non-household caregivers, is a clear reality [9,30]. The National Health Survey was developed in 2014, and subsequently in 2019, showed that in Portugal, about 10% of the population (1.1 million) of people older than 15 years provides informal care [31], underpinning that in the Portuguese context, long-term care depends strongly on informal caregivers, but their needs are not met [15]. Hence, it becomes necessary to analyze the impact of informal care on the health of informal caregivers and their social interactions [9,30].

Our study also reveal that the older people in Portugal lean more towards receiving long-term informal care, related to personal care, from caregivers outside the household who are their children and friends, with a frequency of almost every day, while the older people in Denmark tend to receive more of this care almost every week or almost every month.

Indeed, previous studies conducted in the Portuguese context show a propensity for a high frequency of informal care provided by caregivers who did not reside with older people [9,32]. Generally, this care provision stems from traditional family patterns, characteristic of southern European countries, in which the family has to care for the older people, and not of government institutions [33]. Most of this care is provided mainly by daughters and is most often performed intensively, i.e., daily, with a higher risk of depressive symptoms and burden [34], a situation which contrasts with the Nordic countries’ realities. Even though in the Nordic countries there are more informal caregivers from outside the household, usually the children of older people, the fact is that this care includes other dimensions besides personal care and therefore there are few intensive caregivers [29,35].

The current fact may be related to the differences in political strategies and good social and health governance regarding informal care in Europe [36]. In the case of Nordic countries, in which Denmark is included, there is mostly informal care provided in a non-intensive way, that is, the perspective of playing the role of caregiver is seen as improving human relationships and not so much in knowing how to care in an instrumental way, that is, in providing care related to ADLs and IADLs [7,33].

On the other hand, data from our study demonstrate that older people in Portugal tend to receive long-term informal care from caregivers belonging to the household who are mainly their children or their spouses or partners.

Studies conducted in recent years in Portugal show that in the Portuguese context, there is a high prevalence of informal care provided by caregivers belonging to the household [5,9,11] and that this harms caregivers’ mental health, especially or caregivers older than 50 years, with reported depressive symptoms and low economic and social resources in accessing mental health care [9].

From another perspective, data from our study show that, in general, older people in both countries, as they age, are more likely to receive long-term informal care, referring to personal assistance, either by caregivers outside the household or by caregivers belonging to the household.

This finding is supported by other studies in that the advancement of the ageing process is a factor that provides the need for care from other members of the older people social network, with greater significance for their children [3,30,37].

In addition, data from our study show that older people who do not live with their spouses are more likely to receive long-term informal care (personal care) from caregivers outside the household and are less likely to receive personal care from caregivers belonging to the household.

Recent studies showed that the fact that older people do not live with their spouses is associated with a greater willingness to receive help from caregivers outside the household [30,38]. In some cases, this type of care can be complemented with formal long-term support [38], although this must be framed within the social and health policies adopted by each country.

In turn, the fact that older people do not live with their spouses is associated with a lower likelihood of receiving informal care provided by caregivers belonging to the household, as shown in other studies. Indeed, several studies show that older people who live with their spouses are more likely to receive informal care from caregivers belonging to the household, especially their spouses, and this reality is more evident in Portugal than in Denmark [9]. On the other hand, in the Danish context, there has been an increase in the proportion of older people living alone, which in 2018 was higher than the European average [14].

Other findings from our study demonstrate that older people with high household income are less likely to receive long-term informal care from caregivers outside the household, while people with a higher level of education and a higher household income are more likely to receive care from caregivers in the household.

Concerning studies on income and education level developed in recent years, the results differ. While some authors show that people with a lower income and less education tend to receive more formal care [39], other studies have shown that older people with higher income levels or a higher level of education show a greater ability to receive formal or informal care [30,40]. The results of our study and those found in other studies can be explained by considering several factors, including how social and health policies are implemented to support both formal and long-term informal care, specifically regarding access and financial support [16].

In terms of health, our study shows that older people with two or more chronic conditions are more likely to receive long-term informal care from caregivers outside the household and those with depression are more likely to receive care from caregivers within the household [16].

Indeed, recent studies have shown that the presence of two or more chronic illnesses increases the likelihood that older people will receive long-term informal care due to the process of functional dependency that is set in the process of multiple chronic illnesses [16,30], although they do not specify whether this care is provided by non-household or in-home caregivers, so it is important to conduct further studies to validate this inference.

In turn, the presence of depression increases the likelihood that older people will receive personal care from informal caregivers residing in the home. Such evidence is supported by studies showing that the existence of mental health problems and specifically the case of depression is associated with a decline in the functional abilities of older people and their ability to self-care [41,42]. On the other hand, depression has been identified as a predictor of disability in older people, which increases the need for care assistance [35,36]. However, studies do not detail whether caregivers are external to the household, so more evidence is needed in this regard.

In turn, data from our study showed that older people with one or more limitations in ADLs and IADLs are more likely to receive long-term informal care, referring to personal assistance, either from people outside the household or from home caregivers.

The presence of a limitation in ADLs increases the likelihood of older people receiving long-term informal care by worsening their ability to provide personal care, and studies in both countries support our findings, showing that as the ageing process occurs, the process of dependence on basic activities of daily living (e.g., personal hygiene, dressing/caring for older people, feeding, mobilizing, eliminating, etc.) is becoming a reality [30,34,43].

In turn, the presence of limitations in IADLs because, given the ageing process in approaching the end of life, the ability to deal with the components related to personal management and social and financial resources of older people is diminished. This finding is supported by other studies that show the occurrence of worsening chronic diseases as a factor that increases dependency and thus the possibility of informal care [4,30,32].

Overall, the results of the present study agree with the existing literature, which demonstrates its internal and external validity and adds content to scientific knowledge given the significance of the results obtained. However, it is important to note that the results should be carefully interpreted, and therefore generalized, given the characteristics of the samples.

In addition to exploring the dimension of long-term informal care, this study also contributes to closing the knowledge gap on health data in ageing. The need for health data analysis on ageing is an important issue, as reflected in goal 17 of the Sustainable Development Goals: “*17.18: By 2020, enhance capacity-building support to developing countries, including for least developed countries and small island developing States, to increase significantly the availability of high-quality, timely and reliable data disaggregated by income, gender, age, race, ethnicity, migratory status, disability, geographic location and other characteristics relevant in national contexts*” [44] (p. 24), which demonstrates the need for countries to make age-disaggregated health data available to improve the use and comparison of health data across and within countries and regions and also over time [45,46]. In this context, it is increasingly clear that, throughout ageing, certain factors that predispose people to disease occurrence, increased disability, and the use of health care services vary by age group, and further research in this area is essential [46].

On the other hand, it is important to note that in this study, variables such as the policies implemented for long-term informal care in both countries, as well as the extent of formal care, were not studied, so it is important to analyze them for future research.

Along these lines, it is important to mention that the data from this study can be useful in analyzing how long-term care, specifically informal care, is designed and implemented by health and social support teams. Indeed, as far as health teams are concerned, when nurses are included, the need to know how long-term informal care is designed and implemented will be essential to adjust therapeutic interventions, not only for the older people who need informal help, but also for the informal caregivers themselves.

Nevertheless, the study has some limitations such as (i) the small size of the Portuguese sample as well as that of the Danish sample (even though it was slightly larger than the former, (ii) the low number of older people who responded to the question; and (iii) the impossibility to assess causality because this is a cross-sectional study.

## 5. Conclusions

This study demonstrates the reality of long-term informal care in Portugal and Denmark specifically related to personal care, received by older people.

Ageing in both countries is a similar sociodemographic reality, with a high percentage of older people with chronic disease and dependence.

Nevertheless, Portugal provides long-term informal care to older people who need support in their personal care, either by household or non-household caregivers, which is quite a different reality than that in Denmark.

Therefore, the study demonstrates that it is important to scientifically analyze health and socioeconomic policies in terms of ageing and long-term care, especially when provided by informal caregivers. There is widespread ageing in Europe, but the policies implemented may have equitable perspectives for European citizens, respecting the cultural dimensions and the exercise of freedom of choice in informal care by caregivers and people with disabilities and dependency, safeguarding health promotion.

On the other hand, this study demonstrates the need to look at long-term informal care from both the perspective of the care receiver and the provider. It is important to understand these two realities to adopt public policies that are integrative and that respond to the health and social needs of both the older people who receive care and the caregivers. This issue became even more evident during the 19-COVID pandemic, which highlighted the exposure of informal caregivers to the most vulnerable situations concerning their health status and socioeconomic situation [47].

Finally, it is considered that the future steps can be as follow: (1) conduct a comparative cross-sectional analysis between Portugal and a set of Mediterranean and Nordic countries, which may have similar patterns to the object of the current study and (2) conduct a cross-country and longitudinal analysis to measure and interpret differences related to long-term informal care, namely, between the two countries, which would allow studying how the risk group of the older people coped with the health-related and socioeconomic impact of the recent COVID-19 outbreak.

## Figures and Tables

**Table 1 ijerph-19-10859-t001:** Variables to characterize older people and long-term informal care.

Variables	Categories
**Older people** Age Age class	Years 65–69; 70–74; 75–79; 80–84; 85–89; ≥90 (years)
Gender	Female; male
Marital status	Married and living together with a spouse; registered partnership; married, separated from a spouse ^a^; never married ^a^; divorced ^a^; widowed ^a^
Educational level (Isced-97)	None ^a^; code-1: primary education; code-2: secondary education ^a^; code-3: upper secondary education; code-4: post-secondary education/non-tertiary education ^a^; code-5: the first stage of tertiary education; code-6: the second stage of tertiary education ^a^
Annual household income	Low (<14,844.36 €); medium (≥14,844.36 € to <29,236.09 €); high (≥29,236.09 €)
Country	Denmark; Portugal
Chronic diseases	Less than two; two or more
Depression	Not depressed; depressed
Limitation in ADLs	No limitations; one or more than one limitation
Limitation in IADLs	No limitations; one or more than one limitation
**Long-term informal care received from non-household caregivers**	
Occurrence	Yes; no
Kinship of informal caregiver	Partner/Spouse; Child (son/daughter; stepchild/current partner´s child; son-in-law; daughter-in-law; grandchild); friends; neighbors; other relatives (niece; nephew; ex-spouse/partner)
Frequency	Almost day; almost every week; almost every month ^b^; less often ^b^
**Long-term informal care received from household caregivers**	
Occurrence	Yes; no
Kinship of informal caregiver	Partner/Spouse; Child (son/daughter; stepchild/current partner´s child; son-in-law; daughter-in-law; grandchild); friends; neighbors; other relatives (niece; nephew; ex-spouse/partner)

Notes: ^a^ Categories merged in multivariate analyses, especially due to the low frequencies. ^b^ Categories merged in two-comparison group analyses, especially due to the low frequencies.

**Table 2 ijerph-19-10859-t002:** Sociodemographic, economic, and health characteristics according to the country.

	*N*	Denmark	Portugal	T/χ^2^	*p*-Value	Cohen’s d/φ//V
		*N* = 1878	*N* = 1013			
**Age, years (mean, SD)** **Age, classes (years) (%)**	**2891**	**77.7(1.23)**	**83.0(0.82)**	3.58	*p* = 0.001	0.93 *
65–69	997	33.5	28.6	24.8	*p* = 0.007	0.09 ^+^
70–74	785	25.6	25.2			
75–79	494	17.1	17.5			
80–84	347	12.5	17.3			
85–89	181	7.3	9.2			
≥90	87	4.0	2.2			
Gender (%)						
Female	1520	54.0	58.3	4.8	*p* = 0.006	0.04
Male	1371	46.0	41.7			
Marital status (%)						
Married, living with a spouse	1952	65.0	72.3	126.6	*p* < 0.001	0.21 ^++^
Registered partnership	16	0.2	1.2			
Married, not living with a spouse	28	0.7	1.4			
Never married	104	4.0	2.9			
Divorced	233	10.4	3.7			
Widowed	558	19.6	18.7			
Educational level (ISCED-1997) (%)						
None	92	0.1	9.9	1013.8	*p* = 0.000	0.59 ^+++^
Isced-97 code 1	952	15.9	63.4			
Isced-97 code 2	219	7.1	8.7			
Isced-97 code 3	845	40.8	7.5			
Isced-97 code 4	9	0.3	0.2			
Isced-97 code 5	765	36.0	10.1			
Isced-97 code 6	9	0.2	0.2			
Annual household income (%)						
Low	954	14.1	71.0	987.0	*p* = 0.000	0.58 ^+++^
Medium	954	42.5	23.5			
High	983	43.5	5.5			
Chronic Diseases (%)						
Less than two	430	44.6	26.0	102.6	*p* < 0.001	0.19 *
Two or more	2461	55.4	74.0			
Depression (%)						
Not depressed	660	60.2	51.3	20.5	*p* < 0.001	0.08
Depressed	2331	39.8	48.7			
Limitations in ADLs (%)						
No limitations	2419	93.0	81.2	70.9	*p* < 0.001	0.16 *
One or more than one	472	7.0	18.8			
Limitations in IADLs (%)						
No limitations	2200	84.6	76.2			
One or more than one	691	15.4	23.8	27.7	*p* < 0.0001	0.10 *

Notes: SD: Standard Deviation; Tests for two-group comparison (*t*-test for independent samples (t); chi-square tests (χ^2^); Tests for effect size: Cohen’s d: * small effect (≥0.20); φ-Phi: * small effect (≥0.10); Cramér´s *V*: ^+^ small effect, if df = 2 (≥0.07) if df = 5 or 6 (≥0.05); ^++^ medium effect if df = 2 (≥0.21); if df = 5 or 6 (≥0.13); ^+++^ large effect if df = 2 (≥0.50) if df = 5 or 6 (≥0.22). Source: Survey of Health, Aging and Retirement in Europe- SHARE wave 6 (release 6.1.1), weighted data, *N* = 2891.

**Table 3 ijerph-19-10859-t003:** Long-term informal care characteristics received by older people, relative to personal care, according to country.

	*N*	Denmark	Portugal	χ^2^	*p*-Value	φ
		*N* = 1878	*N* = 1013			
Long-term informal care received from non-household caregivers						
Occurrence (%)						
No	2798	97.3	94.4	11.9	*p* < 0.001	0.06
Yes	93	2.7	5.6			
Kinship of informal caregiver (%)						
-Partner/spouse						
No	2875	99.6	99.6	0.1	*p* = 0.367	0.01
Yes	16	0.4	0.4			
-Child						
No	2837	98.4	96.8	6.1	*p* < 0.001	0.05
Yes	54	1.6	3.2			
-Friends/Other relatives/Neighbors						
No	2842	98.3	97.3	2.9	*p* < 0.001	0.03
Yes	49	1.7	2.7			
Frequency (%)						
-Almost every day						
No	2830	98.5	94.7	23.5	*p* < 0.001	0.09
Yes	61	1.5	5.3			
-Almost every week						
No	2860	98.7	99.6	7.1	*p* < 0.001	0.05
Yes	31	1.3	0.4			
-Almost every month/Less often						
No	2858	98.6	99.4	5.1	*p* < 0.001	0.04
Yes	33	1.4	0.6			
Long-term informal care received from household caregivers						
Occurrence (%)						
No	2752	97.9	87.4	85.2	*p* < 0.001	0.17 *
Yes	139	2.1	12.6			
Kinship of informal caregiver (%)						
-Partner/spouse						
No	2786	98.1	95.2	13.9	*p* < 0.001	0.06
Yes	105	1.9	4.8			
-Child						
No	2867	99.9	94.7	51.1	*p* < 0.001	0.13 *
Yes	27	0.1	5.3			
-Friends/Other relatives/Neighbors						
No	2881	99.8	99.5	0.8	*p* = 0.034	0.02
Yes	10	0.2	0.5			

**Notes:** χ^2^ (chi-square tests); Tests for Effect size: φ-Phi: * small effect (≥0.10); Source: Survey of Health, Aging and Retirement in Europe- SHARE wave 6 (release 6.1.1), weighted data, *N* = 2891.

**Table 4 ijerph-19-10859-t004:** Logistic regression models for the ability to receive long-term informal care.

	Ability to Receive Long-Term Informal Care from Non-Household Caregivers						Ability to Receive Long-Term Informal Care from Household Caregivers					
	Model 1			Model 2			Model 1			Model 2		
	OR	95% IC	*p*-Value	OR	95% CI	*p*-Value	OR	95% CI	*p*-Value	OR	95% CI	*p*-Value
Age, years	1.04	1.01–1.07	*	1.01	0.96–1.05		1.08	1.05–1.12	***	1.06	1.01–1.11	*
Gender												
Female (ref.)												
Male	0.91	0.56–1.47		0.93	0.57–1.51		1.26	0.82–1.93		1.29	0.84–1.98	
Marital Status												
Live with spouse (ref.)												
Does not live with a spouse	1.82	1.09–3.02	*	1.74	1.05–2.90	*	0.17	0.10–0.31	***	0.17	0.13–0.30	***
Educational level												
None-Isced code 2 (ref.)												
Isced code 3- Isced code 6	1.30	0.75–2.26		1.18	0.67–2.08		2.02	1.18–3.48	*	1.96	1.13–3.40	*
Annual Household Income												
Lower (ref.)												
Medium	0.87	0.48–1.56		0.86	0.47–1.56		1.32	0.74–2.36		1.381	0.76–2.50	
Higher	0.22	0.07–0.69	*	0.18	0.06–0.58	**	2.23	1.06–4.67	*	2.13	0.99–4.56	*
Country Denmark (ref.)												
Portugal Country*Age	1.40	0.74–2.63		0.01	0.00–0.79	*	7.35	4.05–13.33	***	0.34	0.00–31.37	
				1.07	1.01–1.13	*				1.04	0.98–1.10	
Number of Chronic Diseases												
Fewer than two (ref.)												
Two or more	2.19	1.20–3.98	*	2.21	1.21–4.04	*	1.54	0.94–2.51		0.94	1.00–2.51	
Depression (Euro-D)												
Not depressed (ref.)												
Depressed	1.13	0.71–1.80		1.12	0.70–1.78		1.60	1.05–2.44	*	1.63	1.07–2.49	*
Limitations in ADLs												
No limitations (ref)												
One or more than one	3.08	1.86–5.09	***	3.10	1.87–5.14	***	6.18	3.95–9.66	***	6.19	3.96–9.69	***
limitation in IADLs												
No limitations (ref.)												
One or more than one	2.90	1.76–4.78	***	2.94	1.78–4.87	***	8.09	5.15–12.71	***	8.26	5.26–12.99	***
	Likelihood Ratio Test:χ^2^(11) = 151.68, *p* < 0.001 Hosmer–Lemeshow Test: χ^2^(8) = 3.70, *p* = 0.883 Nagelkerke R^2^ = 0.218			Likelihood Ratio Test: χ^2^(12) = 156.65, *p* < 0.001 Hosmer–Lemeshow Test: χ^2^(8) = 6.75, *p* = 0.564 Nagelkerke R^2^ = 0.225			Likelihood Ratio Test: χ^2^(11) = 377.64, *p* < 0.001 Hosmer–Lemeshow Test: χ^2^(8) = 9.45, *p* = 0.305 Nagelkerke R^2^ = 0.393			Likelihood Ratio Test: χ^2^(12) = 379.44, *p* < 0.001 Hosmer–Lemeshow Test: χ^2^(8) = 9.43, *p* = 0.308 Nagelkerke R^2^ = 0.395		

Notes: Odds ratios (Ors) with CI (Confidence Intervals 95% in parentheses) are reported: * *p* < 0.05; ** *p* < 0.01; *** *p* < 0.001. Source: Survey of Health, Aging and Retirement in Europe- SHARE wave 6 (release 6.1.1), unweighted data.

## Data Availability

The access to the data collected and generated in the “SHARE” project and used in this research is provided free of charge for scientific use worldwide, subject to European Union and national data protection laws, and the publicly available Terms of Use [22] Access to the data is available online (http://www.share-project.org/data-access.html, accessed on 14 October 2018 after user registration).

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
