# Peer review of "Ageing and Long-Term Informal Care: The Reality of Two Countries in Europe: Denmark and Portugal"

_ijerph, 2022, doi:10.3390/ijerph191710859_

Round 1

Reviewer 1 Report

Dear authors,

It was such a pleasure reviewing your well presented work. The topic addresses in this manuscript is extremely relevant to the presence, considering the aging of the population, particularly for Portuguese reality.

In Methodology section, Table 1 the tittle should be reformulated (includes two sentences with the same information).

In the Discussion section, last paragraph, the limitations presented, particularly the third is not clear about what is not possible to assess (“the impossibility to assess because this is a cross-sectional study”). Also in the future proposals it is not clear why the suggestion of future studies on policies applied in other countries of the Mediterranean area and not the exploration of the measures adopted in the Nordic countries, similar to the object of the current study.

In the Conclusions section it is suggested a reformulation of the content in order to provide a more targeted answer to the study questions initially proposed.

In general the following is suggested:

- English revision of the text by a native speaker;

- To use the expression "older people" instead of "elders”, considering it is the expression used globally for this situation;

- Some errors in the wording of the text should be corrected (ex. extra space between word and comma).

Author Response

Dear Reviewer,

I hope you are well.

We want to thank you for your availability and understanding throughout this manuscript revision process.

Your comments were fundamental and thoughtful in making significant improvements to the manuscript.

Point 1. In the Methodology section, Table 1, the tittle should be reformulated (includes two sentences with the same information).

Response 1: we introduce a correction in the Methodology section, in table 1, whose name is: " Table 1. Variables to characterize older people and long-term informal care”.(Line 290).

Point 2: In the Discussion section, the last paragraph, the limitations presented, particularly the third is not clear about what is not possible to assess (“the impossibility to assess because this is a cross-sectional study”). Also, in future proposals, it is not clear why the suggestion of prospective studies on policies applied in other countries of the Mediterranean area and not the exploration of the measures adopted in the Nordic countries is similar to the object of the current study.

Response 2: corrections in the Discussion section in the final paragraph, specifically: "Nevertheless, the study has some limitations such as i) the small size of the Portuguese sample as well as Danish’s; even though slightly more significant than the former, ii) the reduced number of older people who responded to the question; and iii) the impossibility to assess causality because this is a cross-sectional study. (Lines 1123-1127).

Point 3: In the Conclusions section, a reformulation of the content is suggested to provide a more targeted answer to the study questions initially proposed. 

Response 3:  improve the Conclusion section and reformulate it to provide a more targeted answer to the study questions initially proposed, as can see in these sentences: “This study demonstrates the reality of long-term informal care in Portugal and Denmark, specifically those related to personal care, received by older people. Ageing in both countries is a similar sociodemographic reality, with a high percentage of older people with chronic disease and dependence. Nevertheless, Portugal provides  long-term informal care to older people who need support in their care, either by the household or non-household caregivers, which is quite different from Denmark´s.” (Lines 129-136) and “ Finally, it is considered that the future steps can be as follow: 1) conduct a comparative cross-sectional analysis between Portugal and a set of Mediterranean and Nordic countries, which may have similar patterns to the object of the current study and 2) conduct a cross-country and a longitudinal analysis to measure and interpret differences related to long-term informal care, namely between the two countries, which would allow studying how the risk group of the older people coped with the health-related and socioeconomic impact of recent COVID-19.” (Lines 1152-1159)

Point 4: In general, the following is suggested:- English revision of the text by a native speaker; - Use the expression "older people" instead of "elders”, considering it is the expression used globally for this situation; - Some errors in the wording of the text should be corrected (ex. extra space between word and comma).

Response: English revision was conducted by a proficient user with higher education in the English language; a grammatical revision of the whole text in terms of punctuation and verb tenses; and the expressions "elders" or "elderly" was replaced by the expression "older people".

Reviewer 2 Report

The work is methodologically sound and reports the results of well-organised and conducted cross-cultural research. The comparison between the two countries highlights easily predictable data. The contribution in terms of knowledge is not particularly relevant but corroborates findings from other research. A careful revision of the English text is suggested, which has some errors here and there.

Author Response

Dear Reviewer

I hope you are well.

We would like to thank you for your availability and understanding throughout this manuscript revision process.

Your comments were fundamental and very thoughtful in making significant improvements to the manuscript.

Thus, we inform you that in the light of your comments the following changes have been introduced:

a) English revision by a proficient user with higher education in the English language;

b) grammatical revision of the whole text in terms of punctuation and verb tenses.

Reviewer 3 Report

Dear authors,

thank you for opportunity to review this paper. I believe the manuscript meets expected standards, the background and methods are clearly described, results are well organized and presented. I recommend article for publishing after correcting minor typing errors e.g., care instead of car (line 351).

Author Response

(The authors gave the same response as above.)
